# Fibroblasts in Pulmonary Hypertension: Roles and Molecular Mechanisms

**DOI:** 10.3390/cells13110914

**Published:** 2024-05-25

**Authors:** Hui Zhang, Min Li, Cheng-Jun Hu, Kurt R. Stenmark

**Affiliations:** 1Cardiovascular Pulmonary Research Laboratories, Departments of Pediatrics and Medicine, University of Colorado School of Medicine, Anschutz Medical Campus, Aurora, CO 80045, USA; 2Department of Craniofacial Biology, University of Colorado School of Dental Medicine, Anschutz Medical Campus, Aurora, CO 80045, USA

**Keywords:** adventitial fibroblast, vascular remodeling, hypoxia, inflammation, metabolism, mitochondrial, epigenetics, gene regulation, pulmonary arterial hypertension

## Abstract

Fibroblasts, among the most prevalent and widely distributed cell types in the human body, play a crucial role in defining tissue structure. They do this by depositing and remodeling extracellular matrixes and organizing functional tissue networks, which are essential for tissue homeostasis and various human diseases. Pulmonary hypertension (PH) is a devastating syndrome with high mortality, characterized by remodeling of the pulmonary vasculature and significant cellular and structural changes within the intima, media, and adventitia layers. Most research on PH has focused on alterations in the intima (endothelial cells) and media (smooth muscle cells). However, research over the past decade has provided strong evidence of the critical role played by pulmonary artery adventitial fibroblasts in PH. These fibroblasts exhibit the earliest, most dramatic, and most sustained proliferative, apoptosis-resistant, and inflammatory responses to vascular stress. This review examines the aberrant phenotypes of PH fibroblasts and their role in the pathogenesis of PH, discusses potential molecular signaling pathways underlying these activated phenotypes, and highlights areas of research that merit further study to identify promising targets for the prevention and treatment of PH.

## 1. Introduction

Pulmonary hypertension (PH) is a devastating syndrome associated with high mortality, impacting both pediatric and adult populations across a broad spectrum of disorders [1]. In affected individuals, the pulmonary vasculature undergoes significant cellular and structural modifications within the intima, media, and adventitia. These changes are characterized by enhanced proliferation, resistance to apoptosis, the elevated production of inflammatory mediators, and extracellular matrix (ECM) proteins, as well as metabolic reprogramming [2,3]. Such alterations are also observed in various animal models of PH [4] and are especially pronounced in the pulmonary artery adventitia, evidenced by substantial fibroblast proliferation and the accumulation of macrophages and lymphocytes [5,6,7,8]. The persistent phenotypic transformations in adventitial fibroblasts, maintained over multiple passages in culture ex vivo, are partly mediated through epigenetic mechanisms [5,6,8,9,10]. This body of evidence collectively underscores the significant role of fibroblasts in the pathophysiology of PH.

### 1.1. Fibroblasts in Health and Disease

Fibroblasts were first identified by Rudolf Virchow in 1858 as cells within connective tissue [11] and are a primary cellular component of the fibrous connective tissues dispersed throughout the body [12,13,14]. Together with endothelial and blood cells, fibroblasts are among the most prevalent and widely distributed cell types in the human body [13]. Predominantly of mesenchymal origin, fibroblasts play a crucial role in defining tissue structure through the deposition and remodeling of ECMs and organizing functional tissue networks. Specifically, fibroblasts synthesize ECM proteins including collagens, proteoglycans, elastin, fibronectin, microfibrillar proteins, and laminins, which together constitute the “matrisome” [15,16]. Fibroblasts also actively regulate ECM remodeling through the secretion of lysyl oxidases (LOX), which modulate the cross-linking between collagen and elastin, and through the secretion of metalloproteinases (MMPs and ADAMTs) that specialize in ECM degradation [17]. By doing so, fibroblasts provide essential niches and positional cues to neighboring cells through microarchitectural, biomechanical, and biochemical signals within the ECM. The distinctive ECM composition of each tissue, recognized based on morphological and molecular criteria including gene expression profiling, indicates that fibroblasts from different anatomical locations exhibit considerable variability [18,19]. This diversity underscores their essential role in wound contraction and tissue repair. In 1971, Gabbiani et al. demonstrated that fibroblasts adopt a contractile, myofibroblast phenotype in response to tissue injury [20], a transformation also implicated in fibrosis, which represents chronic, unresolved tissue repair following damage [21]. In the realm of disease, particularly in cancer, fibroblasts significantly contribute to the tumor stroma, depositing ECMs of varied molecular compositions. Beyond their structural role, fibroblasts also secrete a diverse array of growth factors and cytokines that modulate the growth and metastatic potential of resident cells [22,23]. Additionally, their secretion of soluble extracellular signaling molecules is critical during development and in response to injury, providing paracrine signals essential for the differentiation of various cell types [24,25,26,27,28,29].

### 1.2. Challenges in Studying Fibroblast Pathology in Diseases

Fibroblasts exhibit heterogeneity both between and within organs. Early insights into the transcriptional heterogeneity of fibroblasts across different organs were obtained from studies using bulk mRNA isolates from multiple cells, providing molecular perspectives at the organ level [30,31,32]. These studies identified specific molecular markers to differentiate fibroblasts from other cell types, including *VIM*, *PDGFRA* (*Pdgfra*), *FAP* (*Fap*), *FSP1*, and *CD90* (*Thy1*). Subsequent advancements in single-cell RNA sequencing (sc-RNA-seq) have catalogued fibroblasts from virtually all major organs in both mice and humans. Within broader atlas projects, fibroblasts are sometimes referred to by other names, such as stromal cells, mesenchymal stem cells, myofibroblasts [33], and unknown mesenchymal cells [34], highlighting ongoing challenges in cell marker identification and nomenclature. Several studies have specifically focused on the roles of fibroblasts in cancer [35], fibrotic diseases [36], inflammatory diseases [37,38], and development [39]. Notably, Muhl et al. and other researchers showed that the human lung hosts at least four distinct fibroblast populations under normal conditions, compared to three in mice, emphasizing the complexity and functional diversity of these cells in various tissues and conditions [40,41,42].

The phenotypic changes of fibroblasts in response to injury are significant, playing a crucial role in pathological tissue fibrosis as the primary fibrogenic cell type. It was demonstrated five decades ago that fibroblasts transdifferentiate into myofibroblasts, marked by the expression of certain smooth muscle cell (SMC) markers such as Acta2, upon activation by tissue injury or stress [20]. This transition, while a component of the normal wound-healing process, may lead to persistent fibrosis, severely impairing the function of vital organs such as the kidneys, lungs, liver, and heart. The defining feature of fibrosis involves the proliferation of myofibroblasts and excessive deposition of ECMs, resulting in the development of abnormal fibrous connective tissue [22,43]. Noteworthy is the identification of several other cell origins for myofibroblasts, including endothelial cells (via endothelial to mesenchymal transition), macrophages, pericytes, and others [43,44,45], indicating the multifactorial nature of fibrotic disease mechanisms.

### 1.3. Advances in Studying Fibroblast Pathology in Diseases

Thanks to advancements in single-cell technologies, the field of immunology now comprehends the cellular composition of both healthy and diseased tissues in unprecedented detail, and the study of fibroblasts is rapidly evolving. The identification and classification of fibroblast transcriptional subtypes vary across tissues and diseases, reflecting the complexity of these cells. Despite significant progress, a consistent nomenclature for fibroblasts and their subtypes, as well as a conceptual framework for understanding this lineage, remain under development [41]. Recent work by the Buechler and Turley group has employed an integrative single-cell approach to elucidate the organization of this lineage across tissues, diseases, and species, providing valuable insights that could aid in standardizing fibroblast classification [41].

Across various pathologies, including fibrosis and cancer, distinct molecular subtypes of fibroblasts have been shown to promote specific aspects of disease processes, as evidenced in conditions such as rheumatoid arthritis [46], wound healing [47], and various cancers [48]. A recent in vitro study characterized the cellular phenotype and transcriptome of human pulmonary arterial fibroblasts (PAAFs) using bulk RNA sequencing [49]. This was complemented by an in vivo study using single-cell RNA sequencing to explore changes in cellular composition within isolated pulmonary arteries (PAs) from healthy donors and patients with pulmonary arterial hypertension (PAH), providing significant insights into the cellular and molecular changes defining the pathological vascular remodeling process in PH [50].

## 2. Fibroblasts’ Contribution to Pulmonary Vascular Remodeling in PH

Research over the past decade has increasingly supported the concept that the vascular adventitia acts as a biological processing center for the retrieval, integration, storage, and release of key regulators of vessel wall function. The pulmonary vascular adventitia, recognized as the most complex compartment of the vessel wall, comprises a variety of cell types, including fibroblasts, immunomodulatory cells, resident progenitor cells, vasa vasorum endothelial cells, and adrenergic nerves. PAAFs have been identified as exhibiting the earliest, most dramatic, and most sustained proliferative, apoptosis-resistant, fibrotic, and inflammatory responses to vascular stress, thereby playing a pivotal role in the progression of PH [2,51,52,53,54,55,56]. The persistently activated phenotypic changes in adventitial fibroblasts, sustained over multiple passages in culture ex vivo and in the absence of an in vivo environment [8], underscore their significant role. The inability of activated fibroblasts to revert to a quiescent state, even in vitro, is believed to contribute to chronic inflammatory and pulmonary vascular remodeling [52,57].

### 2.1. Disordered Fibroblast Growth in PH

All forms of chronic pulmonary hypertension (PH) are characterized by structural and fibroproliferative changes in both small and large pulmonary arteries through a process known as vascular remodeling. One of the most consistent findings in experimental models of PH, as well as in models of vascular injury and hypertension in systemic circulation, is early and significant adventitial remodeling, characterized by fibroblast proliferation, migration, and differentiation [58,59]. In response to injury and vascular stresses such as overdistension and hypoxia, the adventitial fibroblast is activated and undergoes phenotypic changes, including proliferation. Studies of fibroblasts derived from the adventitia of pulmonary arteries (PAs) suggest a remarkable heterogeneity in their functional characteristics. Fibroblast clones generated from the PA adventitia displayed markedly different proliferative capabilities. However, hypoxic pulmonary hypertensive calves exhibited more than twice as many hypoxia-proliferative fibroblast subpopulations, with a more than two-fold increase in hypoxia-induced DNA synthesis compared to control calf fibroblasts [60]. Chronic hypoxic animal models also demonstrated that PAAFs derived from hypoxic animals exhibit significantly greater growth responses to a variety of growth-promoting stimuli, including hypoxia, than fibroblasts from normoxic animals [61,62]. Several studies showed that PAAFs isolated from idiopathic pulmonary arterial hypertension (IPAH) patients or from hypoxia-induced PH calves (PH-Fibs) exhibited a markedly higher proliferation rate under normoxia in either 10% serum or serum-deprived conditions (serum-free or 0.1% calf serum) over 72 h compared to control fibroblasts (CO-Fibs) using cell count and MTT assays. DNA synthesis ([3H]thymidine incorporation or BrdU incorporation) is also augmented in PH-Fibs [10,63,64,65]. At the molecular level, Hu et al. demonstrated that PAs from human and bovine PH lungs, as well as PH-Fibs cultured from these PAs, exhibit markedly increased expression of growth factors (*FGF1*, *FGF2*, *PDGFb*, and *TGFA*) and apoptosis-resistance genes (*BCL2*, *BCL2L1*, and *BIRC5*). Bovine PH-Fibs also showed significantly increased proliferation genes (*MKI67* and *PCNA*) [66]. Using a kinetic reversible 2-compartment 4k (2T4k) model of the lung 18FLT PET, Ali Ashek et al. showed an increased rate of 18FLT (a marker for cell proliferation) phosphorylation, k3, together with prominent thymidine kinase 1 (TK1) immunostaining in the remodeled PAs of IPAH patient lungs [67]. They also examined the expression of the thymidine metabolism enzymes TK1 and thymidine phosphorylase (TP), as well as a thymidine transporter (equilibrative nucleoside transporter, ENT1), in pulmonary fibroblasts isolated from 9 IPAH patients and 12 donor lungs in culture. IPAH pulmonary fibroblasts exhibited increased expression of ENT1, TP, and TK1 compared to control donor cells and increased expression of thymidine metabolism genes in pulmonary arterial fibroblasts isolated from IPAH patients [67]. The underlying mechanisms controlling this persistently activated, highly proliferative, apoptosis-resistant phenotype of PH-Fibs will be discussed in the following sections of this review.

### 2.2. Proinflammatory Phenotype of Fibroblasts in PH

The persistent accumulation of monocytes and macrophages in the perivascular areas of the pulmonary artery adventitia is a hallmark of PH, consistently observed across various animal models and human cases [68,69,70]. The local immune responses are initiated by both exogenous and endogenous danger signals that engage pattern recognition receptors. Perivascular stromal cells, including fibroblasts, play a crucial role in modulating both innate and adaptive immune responses. As mentioned earlier, the cell states of fibroblasts are dramatically altered in diseased tissues. By interacting with immune cells and influencing the microenvironment towards either inflammation or resolution, fibroblasts serve as active participants in inflammatory diseases [71], fibrosis [72], and cancer [73]. Notably, Li and colleagues demonstrated that hypoxia-induced pulmonary vascular remodeling is characterized by the emergence of a distinct adventitial fibroblast population that exhibits a constitutively activated, or “imprinted”, proinflammatory phenotype capable of inducing the recruitment, retention, and proinflammatory activation of monocytes and macrophages [8]. These proinflammatory fibroblasts generate a microenvironment rich in proinflammatory cytokines such as IL-1beta (*IL1B*) and *IL-6*, macrophage chemoattractant cytokines such as *CCL2*/*MCP1*, *CXCL12*/*SDF1*, and *CCL5*/*RANTES*, macrophage growth and activation factor *GM-CSF*, co-stimulatory molecules like *CD40L*, and the adhesion molecule *VCAM-1* [8]. It was also observed that isolated PAAFs from idiopathic IPAH patients exhibited markedly increased expression of cytokines/chemokines (*CCL2, CSF2*, *CXCL12*, and *IL6*), as well as direct hypoxia-inducible factor (HIF) target genes (*CA9*, *GLUT1*, and *NDRG1*). Hu et al. compared gene expression in endothelial cells, smooth muscle cells, and fibroblasts cultured from PAs of PH patients and PH calves and found that the highest increase in the expression of these genes was observed in PH-Fibs [66]. Fibroblasts’ ability to activate other immune and non-immune cells in PAs will be detailed discussed below.

### 2.3. Fibroblast’s Contribution to Fibrotic Responses in PH

The adventitia, the principal “injury-sensing tissue” of the vessel wall, undergoes many changes during the development of PH. In response to stress or injury, PAAFs are able to proliferate with greater propensity than SMCs and significantly alter their production of ECM molecules. This ECM accumulation can profoundly affect vascular structure and function [59]. The adventitia of remodeled vessels, including those affected by PAH, commonly experience the excessive and progressive deposition of ECM proteins [74,75,76]. A limited number of studies have addressed the role of inflammation in promoting ECM production and remodeling, and thus PA wall thickening, in the context of PH. Inflammation-induced ECM remodeling and PA wall thickening have been proposed given the well-establish theory that PAAFs respond to biomechanical cues by producing inflammatory cytokines such as MCP-1, SDF1, and CCL5, as reviewed above. These inflammatory mediators can subsequently induce PA remodeling and stiffening and thus significantly influence flow dynamics within the vessel and, consequently, right ventricular function [77,78].

In turn, the increased PA pressure also increases PA wall stress, which promotes PAAF profibrotic responses and increases ECM and overall PA stiffness [79]. This concept is supported by Bertero et al.’s study, which cultured human PAAFs on soft or stiff multi-well plates coated with collagen. They found the increased expression of *Col1a1*, *Col3a1*, *CTGF*, *LOX*, *YAP*, and *αSMA* on stiff plates vs soft ones [80]. More interestingly, Mueller et al. recently reported that human female and male PAAFs responded differently to increases in microenvironmental stiffness and serum composition using in vitro experiments employing dynamic poly(ethylene glycol)-alpha methacrylate (PEGαMA)-based biomaterials [81]. Specifically, they found that male PAAFs were more responsive to increases in microenvironmental stiffness, regardless of serum composition. Female PAAF activation followed this pattern only when cultured in serum from younger females (age < 50) or from older females (age ≥ 50) when the serum was supplemented with estradiol. In summary, PAAFs, as both contributors and responders to microenvironmental stiffness, are a critical contributor to the profibrotic phenotype of the PH vessel wall. Future studies will need to consider age and sex differences in the exploration of numerous potential anti- or reverse-ECM remodeling therapies [82].

### 2.4. Role of PH Fibroblasts in Activating Other Cell Types in PH

The pathogenesis of pulmonary arterial hypertension (PAH) involves complex interactions between resident vascular cells (e.g., pulmonary arterial endothelial cells (PAECs), pulmonary arterial smooth muscle cells (PASMCs), and PAAFs) and infiltrating immune cell types, implying a cell-type-specific communication network disrupted in the process of vascular remodeling. Fibroblasts, as the primary stromal cell in the adventitia, have been shown to direct the behavior of immune cells and influence the local immune microenvironment as well as the state of SMCs.

#### 2.4.1. Fibroblasts’ Ability to Induce Macrophage Activation in PH

Macrophages, first identified by Metchnikoff and known for their roles in innate immunity, perform a variety of functions that contribute to tissue homeostasis [83]. Using in vivo imaging and other techniques, recent insights show that fibroblasts and macrophages are often found in close proximity within various tissues, suggesting a potential exchange of growth factors and other signaling molecules that could lead to them influencing each other’s functions [84,85]. 

To determine the cell type(s) responsible for adventitial macrophage activation in PH, El Kasmi et al. exposed naïve primary bone-marrow-derived macrophages (BMDMs) in vitro to intact whole distal pulmonary artery (dPA) explants, adventitia-removed dPA explants, adventitia alone, or conditioned media (CM) generated by ex vivo-cultured dPA adventitial fibroblasts from PH calves and humans [6]. They found that whole dPA explants from calves with hypoxia-induced PH significantly increased the transcription of the macrophage activation markers *Cd163*, *Cd206*, *Il4ra*, and *Socs3* in BMDMs. Removal of adventitia from the dPA explant resulted in a marked decrease of these genes in BMDMs. However, the adventitia alone and CM from both bovine and human PH-Fibs significantly increased the transcription of *Cd163*, *Cd206*, *Socs3*, and *Il4rα* in BMDMs and THP-monocytes. These results indicated that PA adventitia and the adventitial fibroblasts were the cellular source to induce a distinct proinflammatory, profibrotic macrophage phenotype in PH [6]. 

Further, an analysis of genome-wide transcripts and metabolomic profiles in naïve mouse bone-marrow-derived macrophages (BMDMs) treated with media conditioned by adventitial fibroblasts isolated from pulmonary hypertensive (PH-CM) or age-matched control (CO-CM) animals revealed that the PH-CM and CO-CM actively yet differentially influence macrophage transcriptomic landscapes, with the PH-CM triggering pathways related to inflammation, immune response, and metabolism in BMDMs, whereas the CO-CM tended to suppress these pathways. The metabolic status of BMDMs exposed to PH-CM or CO-CM also differed, with the PH-CM enhancing aerobic glycolysis, increasing pentose phosphate pathway activity, altering the tricarboxylic acid (TCA) cycle, and increasing glutamine utilization (Figure 1A) [86]. Distinct upstream regulators, transcriptional regulators, and signaling pathways were identified in the BMDMs treated with CO-CM versus PH-CM (Figure 1B) [86].

While the role of PH-Fibs in macrophage activation is well established, multiple molecular mechanisms could underlie PH-Fib-mediated macrophage activation. El Kasmi et al. demonstrated that PH fibroblasts activate macrophages through paracrine IL6 and STAT3, HIF1, and C/EBPβ signaling to drive the expression of genes in BMDMs previously implicated in chronic inflammation [6] (Figure 1C). Kumar et al. demonstrated a critical role of small extracellular vesicles (sEVs) in PH-CM for macrophage activation because sEVs have proven to be involved in cellular crosstalk in a variety of critical cellular processes, including inflammation, immune response modulation, coagulation, and disease progression [87]. They found that PH-Fibs exhibited increased secretion of sEVs compared with CO-Fibs, and an augmented expression of proinflammatory cytokines/chemokines and metabolic genes in BMDMs in response to sEVs from PH-Fibs was reported. A pharmacological blockade of exosome release from PH-Fibs resulted in the significant attenuation of the proinflammatory activation of BMDMs. Proteomic analyses revealed the significant enrichment of complement and coagulation cascades in sEVs from PH-Fibs, including the augmented expression of the complement component C3. Treatment with siC3-RNA significantly attenuated the capacity of PH-sEVs for the proinflammatory activation of BMDMs (Figure 1C). These results further confirmed fibroblasts’ ability to activate macrophages and indicated that fibroblast-released sEVs can serve as critical mediators of complement-induced perivascular/microenvironmental inflammation in PH [87]. All together, these data are consistent with the hypothesis that the local tissue microenvironment’s composition, including the relative concentrations of metabolites, cytokines, growth factors, oxygen, etc., encompassing signals from proximate cell types, is a decisive factor in determining macrophage metabolism and functionality [6,86,87,88,89,90,91].

#### 2.4.2. Fibroblasts Regulate T-Cell Function in PH

Recent studies have shown that activated fibroblasts can influence not only cells of the innate immune system but also cells of the adaptive immune system, including T-cells [92]. Observational studies have documented various T-cell subsets within the adventitia of pulmonary hypertensive vessel walls. Altered CD4+/CD8+ T-cell ratios have been implicated in PH pathology, contributing to abnormal vascular remodeling of the pulmonary vasculature, indicative of the involvement of T-cells in PH [93]. It has been established that T-helper cells, particularly Th1 and Th17, elicit inflammatory responses through the production of *IL-6*, *IL-2*, *IL-21*, interferon-gamma (*IFNγ*), and tumor necrosis factor (*TNFα*) in PH [94,95]. The impact of these T-helper cells is often exacerbated by Th2 dysregulation, dependent on IL-4 and IL-13 production. Furthermore, a deficiency in the T-regulatory cell (Tregs) population within the hypertensive vessel wall has become increasingly apparent [96]. Tregs, which act to limit the immune response in healthy individuals, typically comprise approximately 5–10% of peripheral CD4+ T-cells [97]. FoxP3+ Tregs, known for their potent immunosuppressive functions, can secrete IL-10 and TGF-β, inhibiting the proliferation of other immune-associated cells, including CD4+, CD8+ T-cells, natural killer (NK) cells, and antigen-presenting cells. An imbalance in the Treg/Th17 ratio has been observed to influence the progression of PH and correlates with the severity and prognosis of the condition [98,99]. Prompted by these observations, Plecita-Hlavata and colleagues sought to examine the effects of activated fibroblasts from the PH vessel wall on T-cell subset numbers and activation status. First, they reported that in vivo CD4+ T-cells located in the dPA of PH calves exhibit a reprogramming of their expression profile towards inflammation, evidenced by the increased expression of IFN gamma, the decreased expression of IL4 and TGFb, and the absence of the Tregs marker FoxP3, assessed using sc-RNA-seq [100]. In vitro experiments exposing isolated human/bovine T-cells to conditioned media from IPAH patients or bovine PH-Fibs (PH-CM) confirmed that human/bovine T-cells differentiated towards a proinflammatory Th1 phenotype, whereas polarization toward the Th2 and suppressive regulatory Tregs+ T subsets was diminished in response to the PH-CM (Figure 1C), [100]. This investigation is crucial for gaining a comprehensive understanding of the immune dysfunction within the adventitial microenvironment associated with PH.

#### 2.4.3. Fibroblasts Direct Smooth-Muscle-Cell-State Transition in PH

SMCs are a key component of the vessel wall, exhibiting remarkable phenotypic plasticity. They can dedifferentiate from a contractile state to a synthetic state in response to inflammation or injury [101]. This transition is a cell-type-specific process and plays a crucial role in cardiovascular diseases, including PH. Crnkovic et al. have reported that pulmonary vascular remodeling is associated with a skewed cellular communication on a single-cell level to define the pathological vascular remodeling process of PAH centered on PAAFs and SMCs [50]. Their most recent findings further uncovered cell-type-specific phenotypic shifts and identified PAAFs as a source of secreted factors that regulate the PASMC state [102]. In this study, Crnkovic and colleagues applied integrative omics and a network-based analysis combined with targeted phenotypic screens to identify common cell-type and cell-state distinct cellular behaviors and examined the bidirectional cell communication between source-matched human PASMCs and PAAFs from healthy controls and IPAH patients. They first demonstrated how a PAAF’s autocrine loop involving complement factors could direct the PAAF towards a disease-like cell state. Subsequently, they identified that PASMCs display a higher intrinsic susceptibility relative to PAAFs in phenotypic responses to external cues such as ECM-mediated or heterotypic cell-to-cell interactions. Ultimately, they concluded that the maintenance and cell-state transition of PASMCs is partly dependent on external cues provided by neighboring PAAFs using a source-matched PASMC and PAAF co-culture model. These results underscore the role of PAAFs as signaling hubs involved in orchestrating heterotypic-cell-state maintenance and transitions [102].

## 3. Mechanisms Underlying Fibroblast Abnormalities in PH

As reviewed above, PAAFs play a critical role in pulmonary vascular remodeling. Therefore, understanding the mechanisms contributing to the sustained activation of PAAFs is essential for elucidating the pathobiology of PH. Various stimuli, such as mechanical strain, hypoxia, inflammation, nutrients, and genetic predisposition can lead to PA vascular cell disfunction [79,103]. Numerous studies have demonstrated that various pathways are involved in the gene regulation process, playing significant roles in controlling development, tissue-specific cell differentiation, cell proliferation, migration, apoptosis, and stress responses. The reported mechanisms underlying fibroblast abnormalities in PH are reviewed in this section.

### 3.1. Role of Dysregulated miRNAs Underlying PH-Fib Abnormalities

Numerous studies have shown that miRNAs are pivotal in the gene regulation process, playing important roles in controlling development, tissue-specific cell differentiation, cell proliferation and migration, apoptosis, and stress responses [104,105,106,107,108]. Thus, it was important to determine whether changes in miRNA expression directly control the activated phenotype of pulmonary fibroblasts in PH. The following paragraphs provide insight into mechanisms governing the expression of one of the miRNAs, miRNA124, and its contribution to the emergence of a persistently activated fibroblast in PH and how they directly contribute to chronic inflammation and remodeling.

Bulk RNA-seq data revealed that PAAFs derived from the remodeled pulmonary arteries of hypoxia-induced PH calves exhibit a striking resemblance to glioblastoma cells that have decreased miR-124. Wang et al. studied PH-Fibs from hypoxic calves and humans with IPAH and provided compelling evidence that the loss of miR-124 is directly involved in the development of activated PH-Fibs, leading to PH [10]. It was thus important to determine the mechanisms contributing to stable decreases of miR-124 in bovine and human PH-Fibs. The ability of an initial stimulus, such as hypoxia, to engage an miRNA circuit could potentially result in stable changes in pulmonary fibroblasts in animals in vivo, which persist even after the hypoxic stimulus is removed. However, different from a report on rheumatoid arthritis synoviocytes, where a hypoxia-induced significant reduction in miR-124 took place [109], the authors did not observe the ability of hypoxia (even prolonged exposure) to induce a stable change in the phenotypes of CO-Fibs toward the activated PH-Fibs or a decrease in miR-124 levels in these cells in vitro, indicating that the activation of the HIF pathway only is insufficient to reduce miR-124 expression and promote the PH-Fib phenotype [10].

To better understand the mechanistic link between miR-124 reduction and the PH-Fib phenotype, Wang and colleagues performed a series of studies and found that miR-124 reduction increases the levels of *HIF-2α* and the RNA-binding protein *PTBP1* (as a direct target of miR-124) in PH-Fibs, and PTBP1 in turn regulates the cell-cycle-related pathway (*Notch1*/*PTEN*/*FOXO3*/*p21* and *p27*), which ultimately contributes to an increase in the proliferative and migratory phenotype of PH-Fibs. Additionally, inhibition of miR-124 upregulates *MCP-1/CCL2* expression, a key chemokine that regulates the migration and infiltration of monocytes/macrophages, leading to increased inflammation. Further, as an RNA-binding protein, PTBP1 controls the alternative splicing of pyruvate kinase muscle isoforms 1 and 2 (PKM1 and PKM2), resulting in an increased PKM2/PKM1 ratio which promotes glycolysis and proliferation in PH-Fibs. On the other side, miR-124 overexpression or PTBP1 knockdown reverses the aerobic glycolysis, rescues mitochondrial abnormalities, and inhibits cell proliferation in PH-Fibs through normalizing the PKM2/PKM1 ratio, at least partly. Pharmacological intervention of PKM2 activity with TEPP-46 and Shikonin, or histone deacetylase inhibition with SAHA and Apicidin, exhibits similar effects [110].

In addition to the effect on glycolysis, low PK enzymatic activity, as would occur when PKM2 is preponderant, has been proposed to divert glycolytic intermediates toward biosynthetic pathways such as the pentose phosphate pathway and serine biosynthesis [111]. Consistent with these studies, steady-state and U-13C-glucose-tracing metabolomics analyses in PH-Fibs (having an increased PKM2/PKM1 ratio) indicated increased pentose phosphate pathway activation and serine biosynthesis and significantly increased purine synthesis [112]. Importantly, it was reported that TEPP-46 or histone deacetylase inhibitors (HDACis) downregulated the substrates of purine de novo synthesis (serine and ribose) and thus purines in PH-Fibs through promoting PKM activity. A synergic therapeutic effect was observed when an HDACi was combined with a well-known vasodilator and a mainstay in the treatment of PH, sildenafil, in correcting metabolic reprogramming and inhibiting the proliferation of PH-Fibs [112]. In summary, the studies by Wang et al. and Zhang et al. reveal how miR-124 regulates PH-Fib phenotypes and provide multiple targets for the development of therapeutic strategies (e.g., HDACi, miR-124 mimics, PTBP1 inhibitors, or PKM2 activators) for the treatment of PH, which are summarized in Figure 2 [10,110,112,113]. 

In addition to the critical role of miR-124 in PA adventitial fibroblasts in PH, Luo Y and colleagues found that activated PAAFs promoted pulmonary vascular remodeling via miR-29a reduction in a chronic hypoxia rat PH model by regulating the levels of a-smooth muscle actin (a-SMA) and ECM collagen. Furthermore, they found that the decrease in miR-29a was regulated by the HIF-1α/Smad3-Associated Pathway. Treatment with an miR-29a-3p mimic inhibited the hypoxia-induced proliferation, migration, and secretion of transforming growth factor-β, endothelin-1, and the platelet-derived growth factor of PAAFs and ameliorated pulmonary vascular remodeling and decreased PA pressure and right ventricle hypertrophy in hypoxic PH rats [114]. Consistent with this finding, significantly decreased levels of miR-29 in the plasma of patients with moderate to severe PAH was reported [115].

### 3.2. Role of Dysregulated HDAC Activity Underlying PH-Fib Abnormalities

The acquisition of stable, functional phenotypic changes likely requires the involvement of epigenetic processes such as those that might occur in response to altered histone modification, DNA methylation, and changes in miRNA expression profiles. The histone-dependent packaging of genomic DNA into chromatin is a central mechanism for gene expression regulation. The expression of miRNA genes, as well as mRNA/protein-coding genes such as inflammatory genes, DNA repair genes, and proliferation genes, is controlled by the degree of acetylation of histone and nonhistone proteins controlled by histone acetyltransferases (HATs) and HDACs [116,117,118]. Several reports have documented HDAC activity changes in fibroblasts in rheumatoid arthritis and juvenile idiopathic arthritis, with a specific increase in HDAC-1 activity [118,119]. In the context of PH, several studies of cells and tissues isolated from the in vivo preclinical models of PH and human PAH demonstrated that the HDAC family is altered in PH, and these alterations are associated with PH pathogenesis through its contribution to histone acetylation, chromatin structure, and gene transcription [49,120]. For example, Li and colleagues first reported the elevated class I HDAC catalytic activity and increased protein levels of HDAC1, HDAC2, and HDAC3 in bovine PH-Fibs, and HDAC inhibitors (HDACis) significantly inhibited the expression of proinflammatory genes and reduced the ability of PH-Fibs to induce monocyte migration and activation [8]. Further, class I HDACs (HDAC1, HDAC2, HDAC3, and HDAC8) were screened in the cardiopulmonary tissues and PAAFs of IPAH patients and healthy donors, and a consistently elevated expression of HDAC1 and HDAC8 in IPAH-PAs and IPAH-PAAFs was reported [64]. In that study, Chelladurai et al. further revealed that isoform-specific silencing and HDACi treatment attenuated IPAH-associated hyperproliferation and apoptosis resistance in IPAH-PAAFs and mitigated PH in a chronic hypoxia rat model [64]. These observations are consistent with previous reports that the selective inhibition of class I HDACs is sufficient to suppress pulmonary vascular remodeling and hypoxic PH [8,121,122].

Recent studies have also explored the link between the abnormal activity of HDACs and the aberrant expression of miRNAs in PH-Fibs. The Wang study demonstrated that HDACis can restore miR-124 levels and decrease its direct target PTBP1 expression toward normal and reverse characteristics of the activated phenotype. Furthermore, using miR-124 expression in human PH-Fibs as a model, Zhang et al. determined that reduced miR-124 gene transcription, but not decreased expression of miRNA processing genes, is responsible for reduced levels of mature miR-124 in human PH-Fibs [113]. The authors also reported a more condensed chromatin structure in the miR-124-1 gene in human PH-Fibs compared to CO-Fibs, which was relaxed by HDACis, reflected in increased levels of the open chromatin mark H3K27Ac and decreased levels of the closed chromatin mark H3K27Me3, as well as increased DNase I sensitivity. Further, direct delivery of histone acetylase (HAT) to miR-124-1 gene regulatory regions using the dCAS9-HAT system increased the activation of miR-124 chromatin structure and gene expression [113]. These findings address the molecular basis for the constitutively activated phenotype of PH-Fibs, which is critical in many aspects of the pulmonary vascular disease process.

### 3.3. The Link between Metabolic Reprogramming and Functionality of Fibroblasts in PH

#### 3.3.1. Metabolic Reprograming in PH-Fibs

In alignment with observations in oncology, studies employing steady-state and U-^13^C-glucose-tracing UHPLC-Mass Spectrometry, as well as steady-state ^13^C-glucose NMR metabolomics technologies, have demonstrated significant metabolic reprogramming in adventitial fibroblasts from the PH vessel wall. These changes, favoring increased aerobic glycolysis over oxidative phosphorylation, encompass enhanced glucose uptake, lactate production, increased one-carbon metabolism, an activated pentose phosphate pathway, incomplete glutamine oxidation, and altered lipid metabolism [65,110]. Additionally, PH-Fibs exhibit mitochondrial abnormalities, including Complex I deficiency, elevated expression of pyruvate dehydrogenase kinase, hyperpolarized mitochondria member potential, increased mitochondrial superoxide production, enhanced fission and mitochondrial fragmentation, and an imbalance in mitochondrial biogenesis and mitophagy [123,124]. 

Pyruvate dehydrogenase (PDH) acts as a crucial enzyme that regulates the entry of glycolysis-produced pyruvate into the mitochondrial tricarboxylic acid (TCA) cycle by converting it to acetyl-CoA, thus governing mitochondrial metabolism. The regulation of PDH phosphorylation by hypoxia, through the HIF1-mediated upregulation of PDH kinase activity, which in turn increases PDH phosphorylation, is well documented [125]. While subjecting CO-Fibs to hypoxia ex vivo is sufficient to diminish mitochondrial respiratory parameters through enhanced PDH phosphorylation, this does not induce the severe metabolic reprogramming or persistent Complex I deficiency as observed in PH-Fibs [123]. This suggests that hypoxia alone and elevated HIF1 signaling in response to hypoxia are insufficient to elicit the activated PH-Fib phenotype [66]. Consistently, a clinical trial indicated that a subset of IPAH patients who have loss-of-function genetic polymorphisms in SIRT3 (sirtuin 3) and UCP2 (uncoupling protein 2) do not respond to dichloroacetate (DCA, a PDH kinase inhibitor), highlighting the involvement of SIRT3 in mitochondrial dysfunction in PH [126]. It has been reported that PH-Fibs with a decreased NAD+/NADH ratio and decreased mitochondrial deacetylase SIRT3 protein levels exhibit increased acetylation levels of general mitochondrial proteins, particularly the acetylation of the SIRT3 target, MnSOD, and a corresponding decrease in MnSOD. Enhancing SIRT3 activity with a combination of the SIRT3 activator Honokiol and the SIRT3 co-factor NAD has been shown to reduce mitochondrial protein acetylation, improve mitochondrial function, and inhibit the proliferation of PH-Fibs [127]. In summary, hypoxia appears to be responsible for part of the metabolic reprogramming of PH-Fibs. We believe that microenvironmental signals, such as lactate, cytokines, and danger signals, either individually or in combination with hypoxia, but not hypoxia alone, are likely critical regulators of cellular metabolic programs that facilitate the activation of inflammatory macrophages displaying similar metabolic alterations [128]

While metabolic reprogramming in PH vascular cells has been well documented, the mechanisms linking metabolic changes to the alteration of gene expression are not well studied. The following section will discuss the role of the metabolic reprogramming in transcriptional regulation (through transcriptional co-repressor C-terminal binding protein-1 (CtBP1)) and epigenetic landscapes in PH-Fibs.

#### 3.3.2. CtBP1 Senses Metabolic Changes and Links to Transcriptional Alterations

CtBP1 is a transcriptional co-repressor. CtBP1’s co-repressor activity is increased in the presence of high free NADH concentrations, a byproduct of glycolytic reprogramming and reduced oxidative phosphorylation in mitochondria. Thus, CtBP1 could serve as a mediator between transcriptional regulation and cellular metabolic status [129,130,131,132,133].

Notably, PH-Fibs (both bovine and human) exhibit persistent elevation in CtBP1 activity even without a hypoxic stimulus, suggesting a stable metabolic reprogramming similar to that seen in cancer cells that drives the activity of transcriptional repressors like CtBP1 [134]. Indeed, knockdown of CtBP1 via siRNA or inhibition of its dehydrogenase activity through MTOB treatment significantly reduces PH-Fib proliferation and normalizes aberrant metabolic phenotypes, similar to the effects observed in cancer therapy where CtBP1 inhibition upregulates the expression of cyclin-dependent kinase inhibitors (*p15* and *p21*) and proapoptotic genes (*NOXA* and *PERP*) [65]. 

While the role of inflammation in PH pathogenesis is critical, its regulation by metabolic reprogramming remains poorly understood [135]. Research concerning the relationship of metabolic reprogramming and inflammation in PH-Fibs has focused on the anti-inflammatory gene heme oxygenase 1 (*HMOX1*, commonly *HO-1*). HO-1 catalyzes the conversion of heme to carbon monoxide and biliverdin, subsequently converted to bilirubin by biliverdin reductase, offering protection against the vasoconstriction and vascular remodeling induced by hypoxia in vitro [136,137,138]. Experiments using HO-1-null mice and transgenic mice with the lung-specific overexpression of HO-1 have highlighted its pivotal role in protecting the right ventricle from hypoxic-pulmonary-pressure-induced injury and early inflammatory changes [139,140]. These findings have promoted investigations into whether targeting CtBP1 at the RNA level via siRNA, or at the activity level through MTOB, or by modulating NADH levels with 2-DG or pyruvate, could potentiate the anti-inflammatory phenotype of PH-Fibs. The findings suggest that small molecules like MTOB or NSC9539 (a CDC25 inhibitor that impedes the CtBP1–adenovirus E1A protein interaction), known for their anticancer properties, might offer new approaches for treating hypoxia-induced PH [141,142]. Preliminary investigations have shown that treating hypoxic mice with MTOB (currently a prohibitively expensive drug) successfully ameliorated vascular remodeling induced by hypoxia [65]. Thus, targeting the transcriptional repressor CtBP1, which bridges metabolic alterations to an activated cellular phenotype, may offer therapeutic benefits, particularly in hypoxia-related PH (group 3). These insights suggest a more integrated approach to understanding vascular cell metabolism, potentially opening new therapeutic avenues for targeting dynamic metabolic reprogramming to attenuate other activated phenotypes of the vascular cells in PH (Figure 2).

#### 3.3.3. The Interplay between Altered Metabolites and Epigenetic Regulation

Moreover, metabolic changes can also alter epigenetic landscapes. Metabolic intermediates serve as both substrates and co-factors for a range of epigenome-modifying enzymes, directly linking metabolic shifts to the chromatin state alterations [143]. For instance, citrate, a direct source of cytoplasmic/nuclear acetyl-CoA (a co-substrate for histone acetylation), is increased, while NAD^+^, a co-substrate for sirtuins (deacetylases), is decreased (relative to NADH). Histone methylation in PH-Fibs is likely altered because S-adenosylmethionine (SAM), the direct methyl donor, and the SAM source amino acids threonine and methionine are increased in PH-Fibs, while inhibitors of the Jumonji C (JmjC)-domain-containing lysine demethylase, such as succinate, fumarate, malate, and 2-hydroxyglutarate, are also increased in PH-Fibs [110,123]. Glucose-6-phosphate dehydrogenase (G6PD), the rate-limiting enzyme in the pentose phosphate pathway, has been found to regulate DNA methylation in sugen-hypoxia and hypoxia-induced PAH mouse models [144,145]. Studies have shown that G6PD expression and activity are increased in various cell types within the pulmonary vascular wall, including fibroblasts from IPAH patients and calves with severe hypoxic PH [65,144,146,147]. Collectively, these findings support the notion that elevated G6PD expression and activity, potentially induced by endothelin-1 and/or hypoxia, inhibit apoptosis and promote the proliferation of various cell types, including PASMCs, ECs, and PAAFs.

Current PAH therapies primarily provide symptomatic relief, targeting vasoconstriction and heart failure but not proliferative vascular remodeling. The bidirectional relationship between epigenetics and metabolomics represents a significant underlying mechanism mediating the pathophysiological modulation of pulmonary vascular remodeling in PAH and warrants further investigation. These alterations underscore the critical role of metabolic adaptations in promoting fibroblast proliferation, apoptosis resistance, and inflammatory activation [148,149].

## 4. Conclusions and Future Directions

In summary, our and other groups’ previous and current findings have demonstrated that pulmonary fibroblasts within the adventitia of pulmonary arteries from experimental animal models of PH and PH patients exhibit a “persistently activated phenotype” compared to fibroblasts from control animals or human subjects. These PH fibroblasts play an active and important role in modulating the microenvironment in pulmonary vessels by promoting and perpetuating the proinflammatory polarization of monocytes/macrophage and T-cells through the secretion of cytokines, chemokines, growth factors, and other molecules. Furthermore, these PH fibroblasts also promote and sustain phenotypic changes of pulmonary vascular stromal cells and vessel remodeling in PH. These studies highlight the important role of fibroblasts in PH pathophysiology via a complex interplay with immune cells and other stromal cells in pulmonary vessels.

As described above, most published results regarding PH fibroblasts are generated from cultured fibroblasts that have been exposed to an in vitro culture environment during the process of establishing the cell population from pulmonary arteries. While a significant number of the gene expression changes found in in vitro cultured fibroblasts have been confirmed in vivo using in situ immunohistochemistry and RNAscope, new technology allows for the examination of genome-wide gene expression changes in vivo in each cell type. Importantly, using single-cell RNA-seq, studies [150] have begun to address the role/change of stromal and immune cells in PH development using PH animal models. However, due to difficulty in isolating the pulmonary arteries in rats and mice, these single-cell analyses were performed in whole lungs, which significantly reduced the power of analysis for the disease organs/tissues, especially the pulmonary vessels. In addition, currently we have no idea if the cells (for example, fibroblasts) found in the lung interstitium of PH animals exhibit similar changes to PH adventitial fibroblasts. Based on the concept that gene expression of even the same cell type is different in different tissues/organs or location, we think that the gene expression of lung interstitial fibroblasts will be different from adventitial fibroblasts. Thus, future studies should explore the single-cell analysis in pulmonary vessels, which will additionally address the heterogeneity of fibroblasts and other cell types. Eventually, a spatial gene expression analysis in situ should be used to determine the crosstalk among the same cell types (same or different clusters) or different cell types. The role of chromatin structure, the activation/binding of transcription factors, and epigenetic regulators in gene expression is well established. We believe that examining chromatin structure, the activation/binding of transcription factors, and epigenetic regulators by technologies such as ATAC-seq, TF-Seq, and histone modification-seq is also critically important to address the molecular mechanism underlying the aberrantly expressed genes in pulmonary vascular cells in chronic PH, as shown in Figure 3, which may uncover new PH therapeutic targets.

## Figures and Tables

**Figure 1 cells-13-00914-f001:**
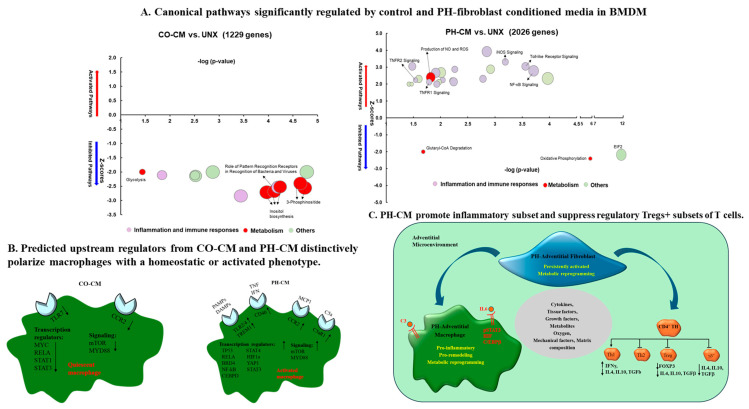
Persistently activated pulmonary artery adventitial fibroblasts regulate the phenotype of adventitial immune cells (macrophage and T-cells) in PH. (**A**) Naïve bone-marrow-derived macrophages (BMDMs) were treated with media conditioned with pulmonary artery adventitial fibroblasts isolated from hypoxia-induced pulmonary hypertensive (PH-CM) or age-matched control calves (CO-CM) or left untreated (UNX). Genes from RNA-seq that are uniquely regulated by CO-CM or PH-CM were analyzed by Ingenuity Pathway Analysis (IPA). The significantly regulated canonical pathways of BMDMs by these genes demonstrated that the CO-CM (**left**) and PH-CM (**right**) have very different effects on regulating BMDM phenotypes. The pathways were selected with *p* ≤ 0.05 and absolute Z ≥ 2. The size of the bubble indicates the number of genes in that pathway. (**B**) IPA predicted upstream regulators in CO-CM- (**left**) and PH-CM-treated (**right**) BMDMs (*p* ≤ 0.05, absolute Z ≥ 2). The upstream regulators are transcripts, proteins, or metabolites that are predicted based on the downstream targets identified by RNA-seq. (**A**,**B**) here were modified from our own publication (Figure 1, Front Immunol 2021 Mar 31:12:640718, reference [86]) (**C**) The complicated microenvironment of the PH pulmonary artery adventitial area regulates the phenotype and function of adventitial macrophages and T-cells.

**Figure 2 cells-13-00914-f002:**
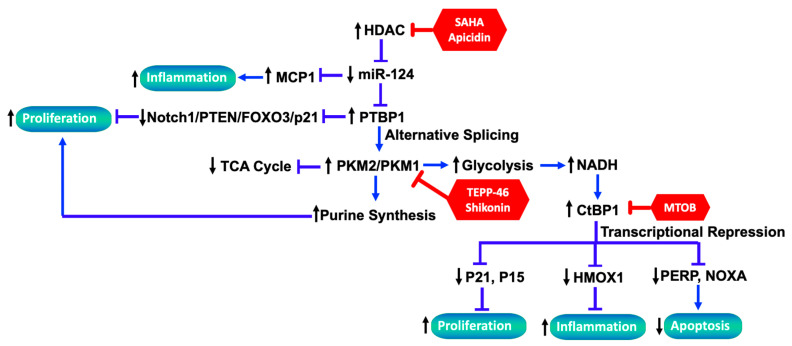
Schematic presentation of mechanisms showing how the HDAC–miR-124–PTBP1/PKM axis controls metabolic reprogramming and the subsequent regulation of phenotypes by the sensor and regulator of glycolysis CtBP1 in PH-Fibs.

**Figure 3 cells-13-00914-f003:**
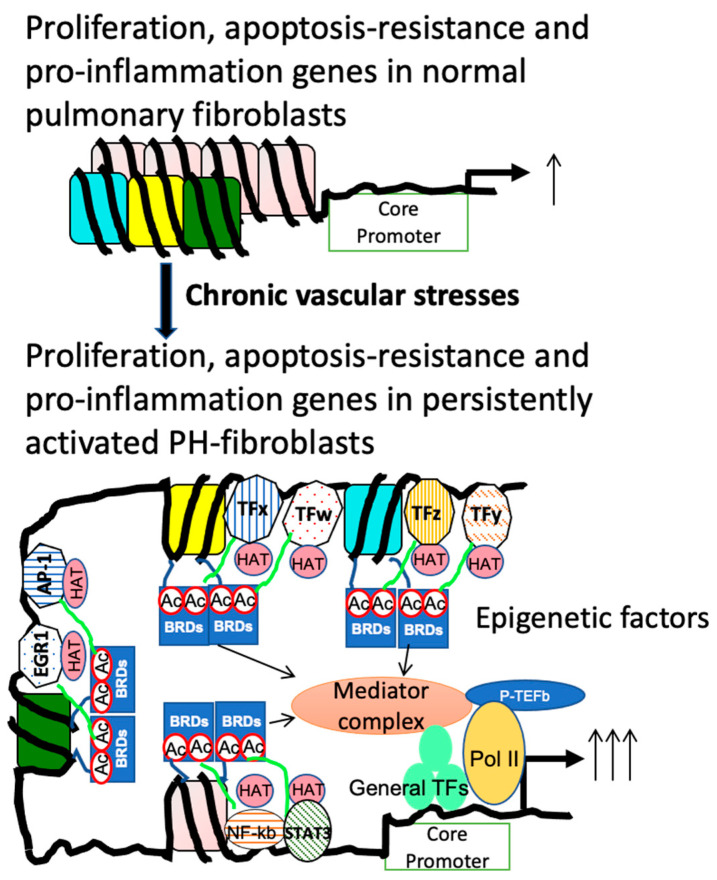
Hypothetical representation of chromatin structure, transcription factors (TFs), and TF co-regulators in normal (**top panel**) and persistently “activated” PH vascular cells (**lower panel**) of genes involved in proliferation, apoptosis resistance, and proinflammation. We posit that the persistently high expression of these genes in PH vascular cells is due to their “open” chromatin structure, allowing the binding of multiple stress-related TFs and pioneer TF(s), which helps maintain an active chromatin structure and high levels of gene expression by recruiting and maintaining high levels of TF co-factors including epigenetic regulators such as HATs, BRDs, and the Mediator Complex (**lower panel**). Abbreviations: Ac, acetylation; EGR1, early growth response 1; p-TEFb, positive transcription elongation factor B; Pol II, RNA polymerase II.

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
