# Peer review of "Fibroblasts in Pulmonary Hypertension: Roles and Molecular Mechanisms"

_cells, 2024, doi:10.3390/cells13110914_

Round 1

Reviewer 1 Report

Comments and Suggestions for Authors

I think that this manuscript regarding the role of fibroblasts in PH is well written and considers all the important aspects of the topic, citing the works already published on the subject in a precise manner.

I therefore believe that it can be accepted for publication

Author Response

Reviewer 1

I think that this manuscript regarding the role of fibroblasts in PH is well written and considers all the important aspects of the topic, citing the works already published on the subject in a precise manner.

I therefore believe that it can be accepted for publication

Response: Thank you very much for your time and effort to review our manuscript and your positive comments.

Reviewer 2 Report

Comments and Suggestions for Authors

This review manuscript presents discussions regarding pulmonary artery adventitial fibroblasts (PAAFs) in the context of pulmonary hypertension (PH), focusing on examining the current understanding of the role of PAAFs during PH pathogenesis and the underlying mechanisms. However, the title suggest a more comprehensive survey of these cells which is currently limited to cell-cell interaction and biomolecular interactions, ignoring mechanical cues dictating the phenotype of PAAF. Summaries of PAAF-related research and suggestions for future directions are also provided.

Pros:

·  Provided introduction to the physiological importance of fibroblasts in general

·  Provided insights into the technical challenges in studying fibroblasts and the corresponding technological advancements that could serve as solutions

·  Provided detailed information on the current understanding of PAAFs’ role during PH pathogenesis, especially regarding the abnormal proliferation, proinflammatory effects, and cell-cell interactions.

·  Provided detailed information on the current understanding of the underlying mechanisms, especially regarding dysregulated miRNAs, epigenetics, and metabolic alterations.

Cons:

·  Recognized the general role of fibroblasts in ECM composition and remodeling, but little information on this aspect is discussed for PAAFs in the context of PH pathogenesis

·  Section 3 provided the cellular and biomolecular mechanisms underlying the abnormalities in PAAF phenotypes, but little information/speculation is provided to explain what pathological cues (other than hypoxia such as mechanical cues) that trigger these abnormalities. i.e., work by Wang A et al. Cells 2021

·  The discussions focus entirely on the cellular and subcellular levels of PH pathogenesis, but missing PAAFs’ interaction with larger scale pathological changes during PH, including tissue level (e.g. vasculature remodeling) and system level (e.g. cardiac and hemodynamic changes)

Author Response

Reviewer 2

This review manuscript presents discussions regarding pulmonary artery adventitial fibroblasts (PAAFs) in the context of pulmonary hypertension (PH), focusing on examining the current understanding of the role of PAAFs during PH pathogenesis and the underlying mechanisms. However, the title suggest a more comprehensive survey of these cells which is currently limited to cell-cell interaction and biomolecular interactions, ignoring mechanical cues dictating the phenotype of PAAF. Summaries of PAAF-related research and suggestions for future directions are also provided.

Response: Thank you very much for your time and effort in reviewing and improving our manuscript. We greatly appreciate your comments and suggestions regarding the coverage of the title and mechanical cues. Please find the addressed comments in point-by point manner as below.

Pros:

  • Provided introduction to the physiological importance of fibroblasts in general

  • Provided insights into the technical challenges in studying fibroblasts and the corresponding technological advancements that could serve as solutions

  • Provided detailed information on the current understanding of PAAFs’ role during PH pathogenesis, especially regarding the abnormal proliferation, proinflammatory effects, and cell-cell interactions.

  • Provided detailed information on the current understanding of the underlying mechanisms, especially regarding dysregulated miRNAs, epigenetics, and metabolic alterations.

Response: Thank you very much for your positive comments.

Cons:

  • Recognized the general role of fibroblasts in ECM composition and remodeling, but little information on this aspect is discussed for PAAFs in the context of PH pathogenesis.

Response: We agree. PAAFs play an important role in dysregulated ECM expression and remodeling, contributing to PH. We have incorporated this information into Section 2 as a new Section 2.3 (Fibroblast’s Contribution to fibrotic responses in PH), Line 183-207. We believe this to be an important addition to the manuscript.

  • Section 3 provided the cellular and biomolecular mechanisms underlying the abnormalities in PAAF phenotypes, but little information/speculation is provided to explain what pathological cues (other than hypoxia such as mechanical cues) that trigger these abnormalities. i.e., work by Wang A et al. Cells 2021

Response: Thank you for your comments and suggestions. We agree that various stimuli, such as mechanical strain, hypoxia, inflammation, nutrients, and genetic predisposition, can directly or indirectly lead to PA vascular cell (including PAAF) dysfunction (Humbert M, 2019, PMID: 30545970; Wang A, 2021, PMID: 34765048). We have incorporated this information into the opening paragraph of Section 3. (Line 327-328). It is also implicated in the new section 2.3 as noted above.

  • The discussions focus entirely on the cellular and subcellular levels of PH pathogenesis, but missing PAAFs’ interaction with larger scale pathological changes during PH, including tissue level (e.g. vasculature remodeling) and system level (e.g. cardiac and hemodynamic changes).

Response: We agree with the reviewer that current manuscript is more focused on the cellular phenotypes and underlying molecular mechanisms regarding the critical role of PAAF in pulmonary hypertension. We have changed the title as “Fibroblasts in Pulmonary Hypertension: Roles and Molecular Mechanisms” (Line 2-3). We believe systems level discussion of fibroblasts and their interactions with other cell in pulmonary and cardiac changes in PH would require a separate manuscript.

Round 2

Reviewer 2 Report

Comments and Suggestions for Authors

All comments were addressed